# A nonmetallic plasmonic catalyst for photothermal $CO_2$ flow conversion with high activity, selectivity and durability

Xueying Wan[1,2,6], Yifan Li [3,6], Yihong Chen[1,2,6], Jun Ma [1,2], Ying-Ao Liu[1,2], En-Dian Zhao[1,2], Yadi Gu[1,2], Yilin Zhao[1,2], Yi Cui [3], Rongtan Li [4], Dong Liu [1,2] ✉, Ran Long [1], Kim Meow Liew [2,5] & Yujie Xiong [1,2] ✉

The meticulous design of active sites and light absorbers holds the key to the development of high-performance photothermal catalysts for $CO_2$ hydrogenation. Here, we report a nonmetallic plasmonic catalyst of $Mo_2N/MoO_2$-$x$ nanosheets by integrating a localized surface plasmon resonance effect with two distinct types of active sites for $CO_2$ hydrogenation. Leveraging the synergism of dual active sites, $H_2$ and $CO_2$ molecules can be simultaneously adsorbed and activated on N atom and O vacancy, respectively. Meanwhile, the plasmonic effect of this noble-metal-free catalyst signifies its promising ability to convert photon energy into localized heat. Consequently, $Mo_2N/MoO_2$-$x$ nanosheets exhibit remarkable photothermal catalytic performance in reverse water-gas shift reaction. Under continuous full-spectrum light irradiation ($3 \, W \cdot cm^{-2}$) for a duration of 168 h, the nanosheets achieve a CO yield rate of $355 \, mmol \cdot gcat^{-1} \cdot h^{-1}$ in a flow reactor with a selectivity exceeding 99%. This work offers valuable insights into the precise design of noble-metal-free active sites and the development of plasmonic catalysts for reducing carbon footprints.

The conversion of $CO_2$ into high-value fuels and chemicals has emerged as a promising approach to address the global energy crisis and mitigate greenhouse gas emissions[1–3]. Among various $CO_2$ utilization routes, the reverse water-gas shift (RWGS) reaction is a critical step to produce $C_1$ feedstock for Fischer-Tropsch synthesis[4]. However, the inherent stability of $CO_2$ necessitates substantial energy inputs for $CO_2$ reduction in traditional thermal catalysis[5,6]. Photothermal catalysis offers a promising alternative to energy-intensive thermal catalysis by harnessing solar energy to actuate reactions[7–9]. This approach

can drive catalytic reactions under mild conditions and reduce energy consumption effectively. The remarkable potential of photothermal catalysis in improving the efficiency of chemical processes and solar energy utilization will promote advancements in energy, sustainability and materials science[10,11].

Photothermal catalysis toward practical applications requires the development of efficient catalysts by precisely designing and constructing catalytically active sites. Generally, the adsorption and activation of reactants on the active sites together with the

[1]Hefei National Research Center for Physical Sciences at the Microscale, Collaborative Innovative Center of Chemistry for Energy Materials (iChEM), Key Laboratory of Precision and Intelligent Chemistry, School of Chemistry and Materials Science, National Synchrotron Radiation Laboratory, School of Nuclear Science and Technology, University of Science and Technology of China, Hefei 230026 Anhui, China. [2]Sustainable Energy and Environmental Materials Innovation Center, Nano Science and Technology Institute, Suzhou Institute for Advanced Research, University of Science and Technology of China, Suzhou 215123, China. [3]Vacuum Interconnected Nanotech Workstation, Suzhou Institute of Nano-Tech and Nano-Bionics, Chinese Academy of Sciences, Suzhou 215123, China. [4]State Key Laboratory of Catalysis, Dalian Institute of Chemical Physics, Chinese Academy of Sciences, Dalian 116023, China. [5]Centre for Nature-Inspired Engineering, Department of Architecture and Civil Engineering, City University of Hong Kong, Kowloon, Hong Kong, China. [6]These authors contributed equally: Xueying Wan, Yifan Li, Yihong Chen. ✉e-mail: dongliu@ustc.edu.cn; yjxiong@ustc.edu.cn

light-harvesting process are crucial to photothermal catalysis. Noble-metal/oxide composite nanostructures have been commonly employed as effective constituents for photothermal $CO_2$ hydrogenation[12–15]. In such systems, noble-metal nanoparticles with an localized surface plasmon resonance (LSPR) effect can enhance light utilization and generate hot carriers for chemical reactions[10,13]. Meanwhile, oxide components provide oxygen vacancies ($V_o$) to facilitate $CO_2$ adsorption and activation[16,17]. For example, $Au/TiO_2$ is one typical photocatalyst with LSPR effect[18–20]. Fan et al. proved that hot electrons generated by LSPR can promote the formation of oxygen vacancies in $Au/TiO_2$ catalyst, facilitating the adsorption and activation of $CO_2$[19]. Sastre et al. demonstrated that plasmonic $Au/TiO_2$ nanostructure could drive photothermal RWGS reaction with a CO generation rate of 429 mmol·$g_{Au}^{-1}$·$h^{-1}$ (13.4 mmol·$g_{cat}^{-1}$·$h^{-1}$) under 14.4 sun irradiation[20]. The photothermal catalysts composed of noble metals and oxides were usually synthesized via a multi-step process[21]. In addition to noble metal nanostructures, some semiconductor nanostructures also exhibit LSPR in visible-NIR regions. The adaptable surface of semiconductors can be engineered to provide plentiful active sites for photothermal reactions, such as the creation of oxygen vacancies in $WO_3$ and $MoO_3$[22–24]. However, these catalysts still require the involvement of noble metals to activate $H_2$ and boost their catalytic activity, which increases the intricacy of the catalyst synthesis process[14,15,25]. For instance, Mo-doped $Pt/WO_y$ nanostructures were formed using a multi-process method, which could achieve a CO production rate (3.1 mmol·$g_{cat}^{-1}$·$h^{-1}$) at 140 °C benefiting from the incorporation of Pt nanoparticles[14]. To fulfil the requirements for application-oriented photothermal $CO_2$ conversion, it is imperative to develop rational design strategies and feasible fabrication techniques for nonmetallic plasmonic catalysts along with high activity and durability under mild conditions.

Here, we report a $Mo_2N/MoO_{2-x}$ nonmetallic plasmonic catalyst with two specific active sites for highly efficient photothermal $CO_2$ hydrogenation. Implementing a facile one-step annealing treatment under ammonia, $MoO_3$ precursor can be transformed into $Mo_2N$ with dispersed $V_o$-rich $MoO_{2-x}$ nanoclusters. The obtained $Mo_2N/MoO_{2-x}$ catalyst shows excellent activity, selectivity and durability for photothermal RWGS reaction. In such a system, the optimal catalyst can produce CO with a yield rate as high as 355 mmol·$g_{cat}^{-1}$·$h^{-1}$ (selectivity > 99.9%) for a continuous 168 h test in a flow reactor, indicating a great potential for practical application. Combining in-situ spectroscopic characterizations and density functional theory (DFT) calculations, we elucidate the in-depth mechanism of plasmonic $Mo_2N/MoO_{2-x}$ catalyst for photothermal RWGS reaction. In the RWGS reaction, two types of active sites, i.e., N atoms and oxygen vacancies, work synergistically, enabling the adsorption and activation of $H_2$ and $CO_2$ molecules simultaneously. Owing to the synergism of dual active sites, the activation energy barrier can be reduced significantly. The strong LSPR effect in the catalyst also plays a vital role in promoting the RWGS reaction and boosting energy conversion efficiency by locally converting photon energy into thermal energy. The LSPR-induced photothermal catalysis utilizing nonmetallic plasmonic catalysts offers a viable approach to optimize localized heat management, catalytic property and cost performance toward practical applications[7,21,26].

## Results and discussion

### Synthesis and characterization of $Mo_2N/MoO_{2-x}$ catalyst

The $Mo_2N/MoO_{2-x}$ nanosheets are synthesized by high-temperature ammoniation from $MoO_3$ precursor (Fig. 1a). The resulting $Mo_2N/MoO_{2-x}$ samples are designated as MNO-450, MNO-550 and MNO-650, respectively, corresponding to their annealing temperatures. Scanning electron microscopy (SEM), high-resolution transmission electron microscopy (HRTEM) and energy-dispersive X-ray spectroscopy (EDS) mapping analysis demonstrate the well-shaped nanosheet morphology, crystallographic structure and uniform element distribution of MNO-550 (Fig. 1b–d and Supplementary Fig. 1a). The lattice spacing of 0.24 nm can be assigned to the (111) plane of $Mo_2N$, and that of 0.28 nm corresponds to the (101) plane of $MoO_2$. Additionally, a considerable number of pore structures are discernible with an approximate diameter of ~3 nm, which is consistent to the result from Brunauer-Emmett-Teller (BET) measurements (Supplementary Fig. 1b, c). By varying the annealing temperature, we can tune the composition ratio of $MoO_{2-x}$ and $Mo_2N$. X-ray diffraction (XRD) shows that the diffraction peaks for cubic $Mo_2N$ become more pronounced as the annealing temperature increases (Fig. 1e), indicating the positive correlation of $Mo_2N$ content with annealing temperature.

To look into the electronic structures and chemical states of $Mo_2N/MoO_{2-x}$, X-ray photoelectron spectroscopy (XPS) and X-ray absorption near-edge structure spectroscopy (XANES) are employed to examine different Mo-based catalysts[27,28]. The coordination information for Mo atoms is provided by the Mo K-edge of MNO-550 in the Fourier transform extended X-ray absorption fine structure (FT-EXAFS) spectra (Supplementary Fig. 1d), revealing the co-existence of Mo-N and Mo-O. XPS characterization shows that the intensity of N 1s component increases with the annealing temperature (Fig. 1f), consistent with the XRD results. Meanwhile, there is a discernible drop in the intensity of Mo 3p component at higher binding energy (398.7 and 416.2 eV), indicating a gradual reduction of $MoO_{2-x}$ by ammonia when increasing the annealing temperature (Supplementary Table 1). The adsorption edge position of MNO-550 nanosheets is observed to locate between those of reference samples in the XANES spectra (Fig. 1g), suggesting that it possesses composite valence states. As a result, a series of molybdenum-based catalysts ($MoO_3$, $MoO_{2-x}$, and $Mo_2N/MoO_{2-x}$) with tunable component ratios of nitride and oxygen vacancy have been obtained by the one-step annealing method.

### Photothermal $CO_2$ hydrogenation performance of $Mo_2N/MoO_{2-x}$ catalyst

Prior to photothermal performance assessment, we first examine the ability of effectively utilizing light for photothermal catalysis. The UV-vis-NIR absorption spectra (Fig. 2a) show that $Mo_2N/MoO_{2-x}$ exhibits an obvious broad-spectrum light adsorption capacity (250–1800 nm) in contrast to its counterparts (i.e., $Mo_2N$ and $MoO_2$ components, which are reported as nonmetallic plasmonic materials)[24,29]. The enhanced LSPR performance of $Mo_2N/MoO_{2-x}$ can be attributed to its distinctive nano-architecture. On the one hand, the broad grain size distribution ranging from 15 to 270 nm (Supplementary Fig. 2) can result in the overlapping and widening of plasmonic charaters[30,31]. On the other hand, a large number of interfaces exist between $Mo_2N$ and $MoO_{2-x}$, and their interfacial interaction will influence the strength of LSPR. Consequently, a larger photocurrent response was observed (Supplementary Fig. 3) in $Mo_2N/MoO_{2-x}$ nanosheet owing to the synergistic contribution of plasmonic $Mo_2N$ and the oxygen vacancies in $MoO_{2-x}$ c[23,24]. In addition to the effective promotion of photogenerated charge kinetics, it is noteworthy that the accumulation of hot electrons excited by LSPR effect at the interface can rapidly raise the surface temperature of MNO-550 up to 250 °C without external heat supply (Supplementary Fig. 4)[32]. The LSPR generated in the catalysts strengthens light utilization efficiency and converts photon energy into thermal energy locally, thereby facilitating catalytic reactions[9]. Finite-difference time-domain (FDTD) simulations were further conducted to confirm the existence of a pronounced LSPR effect in $Mo_2N/MoO_{2-x}$ (Fig. 2b). An increase in the strength of local electric field is observed in $Mo_2N/MoO_{2-x}$, approximately 3 times higher than those of individual component materials[33,34].

Upon acknowledging the LSPR effect, we proceed to investigate the photothermal catalytic properties of catalysts. The performance of Mo-based catalysts for RWGS reaction is evaluated by feeding mixed gas of $CO_2$ and $H_2$ (1:1) under 3.0 W·$cm^{-2}$ full-spectrum illumination without external heat supply. The predominant product of these

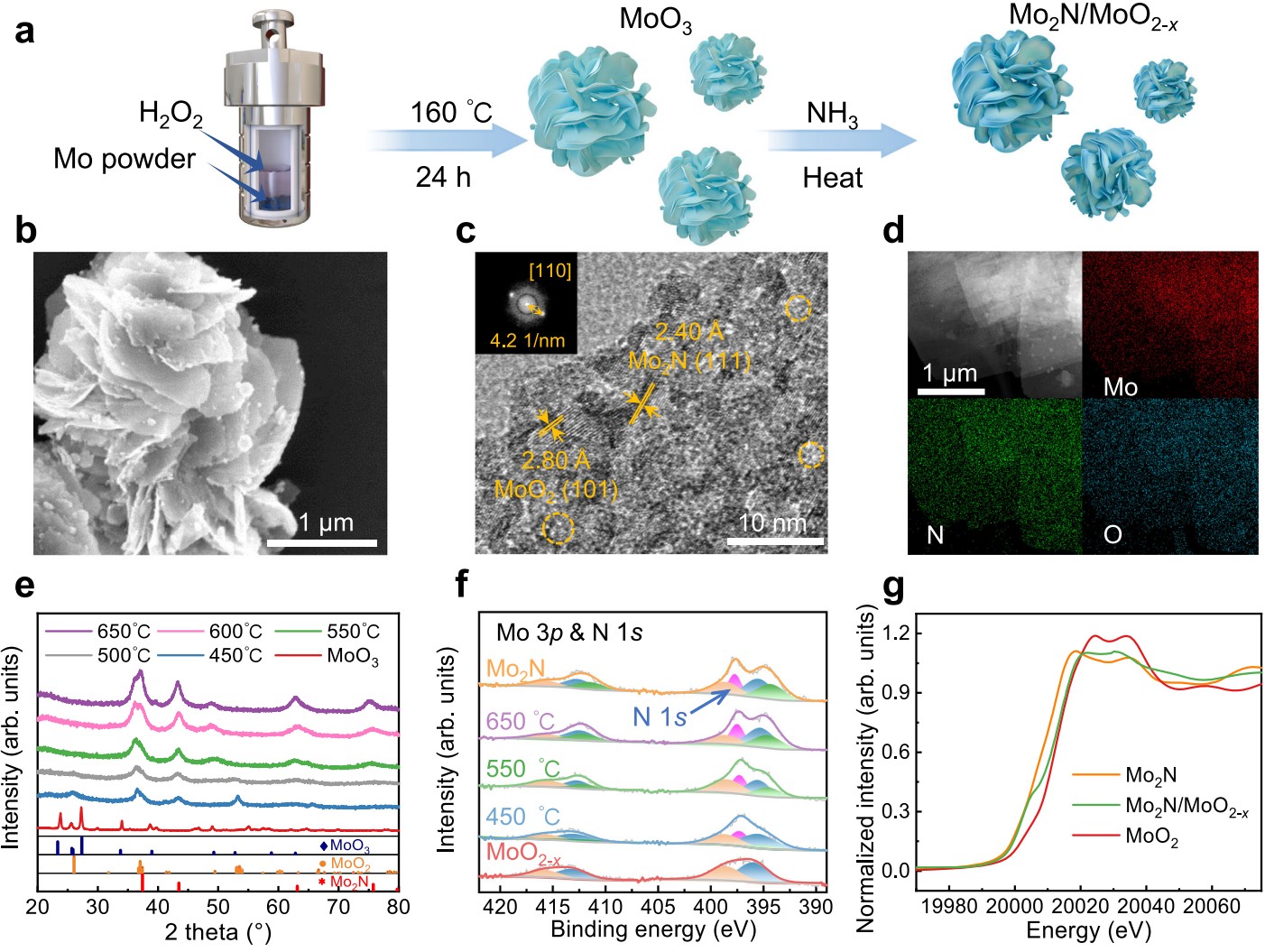

**Fig. 1 | The synthesis method and structural diagrams of catalysts. a** The synthesis route of $Mo_2N/MoO_{2-x}$. **b** SEM image, **c** HRTEM image and (**d**) element distribution of MNO−550. **e** XRD patterns of $MoO_3$ and $Mo_2N/MoO_{2-x}$ samples at different annealing temperatures. **f** Mo $3p$ and N $1s$ XPS spectra of $Mo_2N$, $MoO_{2-x}$ and $Mo_2N/MoO_{2-x}$ samples at different annealing temperatures. The orange, blue, green and magenta peaks are attributed to $MoO_3$, $MoO_2$, Mo $3p$ and N $1s$ of $Mo_2N$ components, respectively. **g** XANES spectra of $Mo_2N$, $MoO_{2-x}$ and MNO−550 at Mo K-edge.

Mo-based catalysts is CO with ~99% selectivity. Notably, the catalytic efficiency of $Mo_2N/MoO_{2-x}$ samples is substantially higher than that of $MoO_3$ precursor, commercial $MoO_{2-x}$, MoN and $Mo_2N$ when subjected to identical reaction conditions (Fig. 2c). Among the catalysts, MNO-550 obtains the optimal photothermal RWGS activity in a batch reactor attributing to its optimized synergic component (Supplementary Figs. 5, 6). The CO yield rate of MNO-550 is 389 mmol·$g_{cat}^{-1}$·h$^{-1}$ (selectivity > 99%) for 20 min, which is 208 times and 130 times higher than that of $MoO_{2-x}$ and $Mo_2N$, respectively. In contrast, without in-situ constructed interfacial interactions of $Mo_2N$ and $MoO_{2-x}$, the synergistic enhancement of catalytic activity cannot be observed in the mechanically mixed samples (Supplementary Fig. 7).

Impressively, during continuous testing in a flow reactor with the flow rate of 20 SCCM (mL·min$^{-1}$) (Supplementary Fig. 8), MNO-550 remains highly active and stable for at least one week, achieving an average CO production rate at 355 mmol·$g_{cat}^{-1}$·h$^{-1}$ and 13.1% $CO_2$ conversion rate (selectivity > 99.9%) (Fig. 2d and Supplementary Fig. 9). The CO generation rate has merely decreased by 12% after 190 h. In successive light on and off conditions, the catalytic performance of MNO-550 also remains steady (Supplementary Fig. 10). After a long period of catalytic process, the morphology and phase of MNO-550 remain consistent (Supplementary Fig. 11), which proves the stability

of this catalyst. As previously mentioned, LSPR-induced photothermal effect elevates surface temperature of MNO-550 up to 250 °C (Supplementary Fig. 4). In contrast, to attain the same $CO_2$ conversion rate in thermal catalysis, it would require a dramatically higher reaction temperature at ~500 °C for MNO-550 (Fig. 2e and Supplementary Fig. 12a) even at lower flow velocity. The reduced temperature requirement for highly efficient RWGS reaction in our system also signifies that the introduction of LSPR effect not only induces a localized photothermal effect, but also promotes the charge dynamics to actuate the reactions. Meanwhile, the surface temperature and CO generation rate of the catalyst are positively correlated with light intensity (Supplementary Fig. 12b), which indicates the excellent photothermal conversion ability of $Mo_2N/MoO_{2-x}$ nanosheets[7]. In addition, the energy conversion efficiency of photothermal catalysis in our system has a significant advantage over thermal catalysis for RWGS reaction (Supplementary Table 2). Specifically, the conversion efficiency of thermal energy to chemical energy by MNO-550 in photothermal catalysis is 4 ~ 5 times higher than that in thermal catalysis, while the required reaction temperature can be reduced by 230 ~ 300 °C in photothermal catalysis compared to thermal catalysis. Consequently, our $Mo_2N/MoO_{2-x}$ nanosheets exhibit excellent performance in terms of both stability and activity even under mild

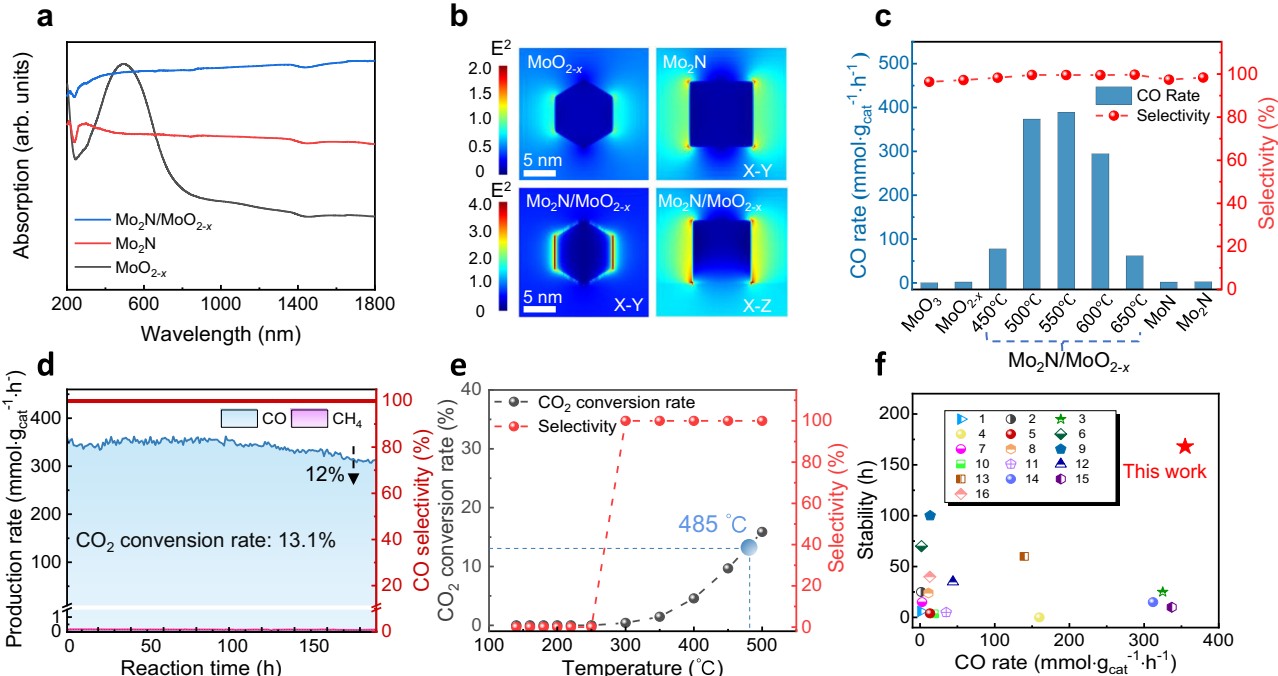

**Fig. 2 | Photothermal CO₂ hydrogenation performance of Mo₂N/MoO₂₋ₓ catalyst. a** UV-vis-NIR absorption spectra and (**b**) spatial distribution of electric fields enhanced by LSPR effect under the excitation wavelength of 497 nm of MoO₂₋ₓ, Mo₂N and Mo₂N/MoO₂₋ₓ. **c** The production rate and selectivity of CO evolution in a batch reactor by Mo-based catalysts under 3 W·cm⁻² full-spectrum light irradiation. **d** Long-term stability of MNO-550 in a flow reactor under 3 W·cm⁻² full-spectrum light irradiation with the flow rate of 20 SCCM (H₂/CO₂ 1:1). **e** The CO₂ conversion rate and CO selectivity of MNO−550 at various temperatures in thermal catalysis.

**f** The comparison of catalytic performance with state-of-the-art catalysts for photothermal catalytic RWGS reaction, and the specific test parameters, such as light intensity and stability test time, are displayed in Supplementary Table 3[12,25,27,33,35–44]. Serial numbers 1–16 represent Pt/HₓMoO₃₋ᵧ[24], CuNi/CeO₂[28], Ni₃N[44], Au/TiO₂ (DP)[18], Au/TiO₂[20], Black In₂O₃[36], Mo₂NH₋ₓ[27], Fe₃O₄[37], Ni₁₂P₅/SiO₂[38], Pd@Nb₂O₅[12], Cu/2D-Si[39], Ni@SiO₂[40], CF-Cu₂O[41], Ru/Mo₂TiC₂[42], Ga-Cu/CeO₂[43] and TiN@TiO₂@In₂O₃₋ₓ(OH)ᵧ[33], respectively.

conditions, as illustrated in Fig. 2f, in comparison with the state-of-the-art oxide or nitride catalysts for photothermal catalytic RWGS reaction (Supplementary Table 3)[12,25,27,33,35–44].

## Spectroscopic characterizations for catalytic mechanism

To elucidate the catalytic mechanism, comprehensive characterizations are performed to gain insight into the origin of remarkably enhanced photothermal driven RWGS activity on Mo₂N/MoO₂₋ₓ. As previously reported, Vₒ often acts as the active site for the adsorption and activation of CO₂[23,25,36]. We thus examine whether the high-temperature ammonization annealing can introduce oxygen vacancies in our catalysts. The electron paramagnetic resonance (EPR) spectra in Fig. 3a show a response at g = 2.0003 indicating molybdenum-oxygen vacancies embedded in the lattice[45]. Specifically, the strongest signal is detected from MoO₂₋ₓ, while MNO-550 exhibits the signal implying its moderate number of oxygen vacancies among the three Mo₂N/MoO₂₋ₓ samples. In addition, XPS can also be applied as an effective technique to investigate Vₒ[25,46]. The intensities of peaks at 531.5 eV attributed to Vₒ exhibit the same variation tendency as EPR spectra (Fig. 3b). It turns out that the concentration of oxygen vacancies is determined by the annealing temperature. As such, we can modulate the proportion of MoO₂₋ₓ in Mo₂N/MoO₂₋ₓ by adjusting the annealing temperature.

We also look into the possible changes of catalysts along with photothermal RWGS reactions. After the reactions, a new peak emerges at 533.3 eV in the XPS spectra of MoO₂₋ₓ, MNO-450 and MNO-550 (Fig. 3c), which can be assigned to adsorbed oxygen species[25]. Impressively, these three samples still possess adequate oxygen vacancies after the reactions. The C 1s spectra of Mo-based catalysts before and after the reactions are also compared (Supplementary Fig. 13), suggesting the strong adsorption of carbon oxide species on the oxygen vacancies[45,47]. Meanwhile, temperature-programmed

desorption (TPD) profiles combined with in-situ XPS in CO₂ atmosphere indicate that MNO-550 exhibits a moderate adsorption capacity among the three Mo₂N/MoO₂₋ₓ samples (Supplementary Figs. 14 and 15), which is consistent with XPS results. A positive correlation can be observed between the amount of adsorbed CO₂ species and the concentration of oxygen vacancies. This correlation suggests that oxygen vacancies primarily function as the active sites for CO₂ adsorption and activation on the Mo₂N/MoO₂₋ₓ catalyst surface.

To gain a comprehensive understanding of the photothermal driven CO₂ reduction reaction process, in-situ spectroscopic characterizations are employed to investigate the dynamic change of reaction intermediates and the light-induced charge transfer path on the catalyst surface. In-situ EPR spectroscopy results show that the typical signal at g = 2.0003 increases significantly under illumination (Supplementary Fig. 16). This feature proves the presence of photo-generated oxygen vacancies during RWGS reaction, which is beneficial for the catalysis reaction process. In-situ near-ambient-pressure XPS (NAP-XPS) is further performed in the reaction atmosphere (Fig. 3d, e). When the surface of MNO-550 is exposed to 0.3 mbar CO₂ and 0.3 mbar H₂ without illumination, an identifiable C 1s peak emerges at 285.6 eV. This new peak is attributed to the C = O configuration, considered as a surface reaction intermediate. Simultaneously, two O 1s peaks at 533.3 and 537.2 eV, ascribed to the adsorbed C = O species and gaseous CO₂, respectively, can also be recognized[47]. Upon illumination, the N 1s peak encounters a shift toward lower binding energy by 0.2 eV, while the Mo 3p doublet peaks maintain their original position, indicating the migration of photogenerated electrons onto surface N sites[27]. At the same time, the intensity of C 1s peak at 285.6 eV (C = O peak) increases considerably (Fig. 3e), suggesting that the activation of CO₂ is promoted by the strengthened local electric field under light illumination[47]. When light is turned off, the N 1s peak returns to the

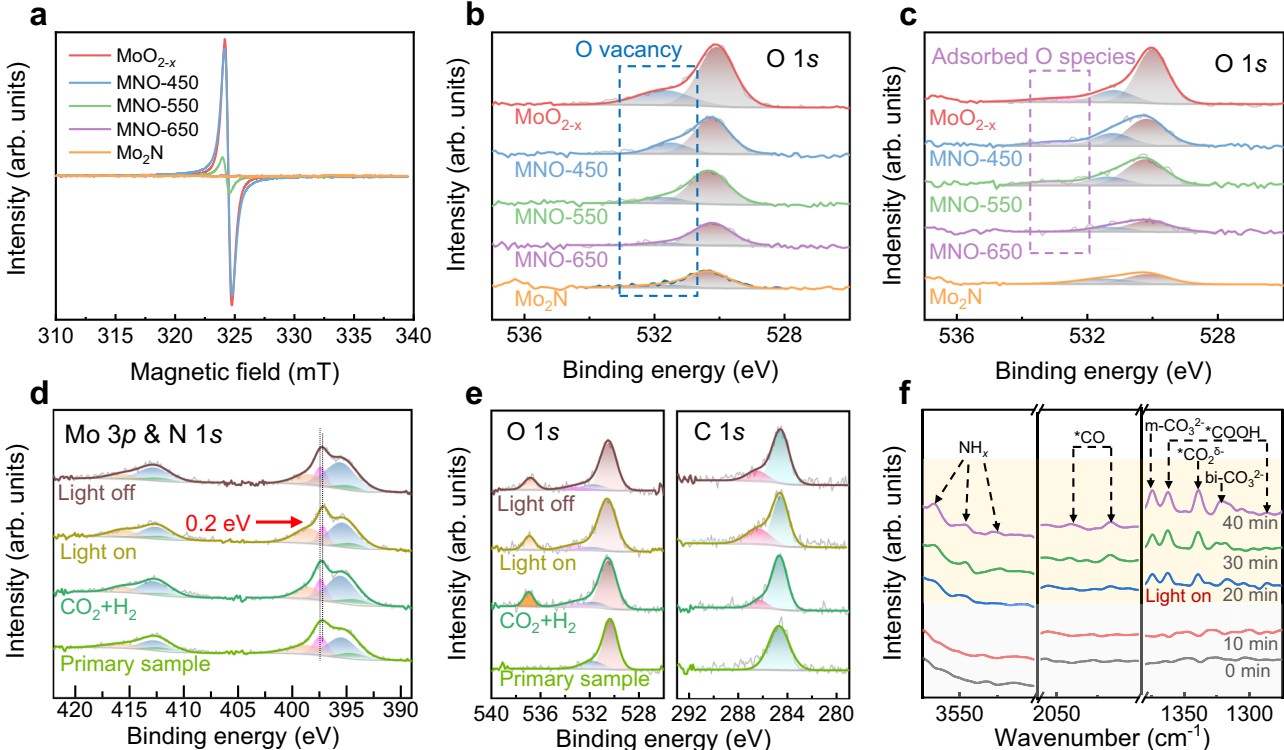

**Fig. 3 | Spectroscopic characterizations for catalytic mechanism. a** EPR spectra and (**b**, **c**) O 1*s* XPS spectra of different samples (**b**) before and (**c**) after the reaction. **d**, **e** In-situ NAP-XPS spectra of (**d**) Mo 3*p* & N 1*s* and (**e**) O 1*s* and C 1*s* for MNO−550 under full-spectrum light irradiation in 0.3 mbar $CO_2$ + 0.3 mbar $H_2$. **f** In-situ DRIFTS measurements of MNO−550 under full-spectrum light irradiation in a mixed atmosphere of 50% $CO_2$ and 50% $H_2$. In Fig. 3b, c, e, the orange, purple, blue and brown peaks are attributed to gaseous $CO_2$, adsorbed oxygen species, oxygen vacancy and metal oxygen, respectively. In Fig. 3d, the orange, blue, green and magenta peaks are attributed to $MoO_3$, $MoO_2$, Mo 3*p* and N 1*s* of $Mo_2N$ components, respectively.

pristine position due to the consumption of photogenerated electrons by reactions, and the strength of C = O peak decreases illustrating the decline of reaction rate without illumination.

In-situ diffuse reflection Fourier transform infrared spectroscopy (DRIFTS) is also conducted to detect the evolution of surface species. There are no evident peaks observed within the designated measurement range in a mixed atmosphere of $CO_2$ and $H_2$ during the initial 20 min period without illumination (Fig. 3f). Once exposed to full spectrum illumination, a series of peaks appear promptly and reach a consistent state within 20 minutes. The peaks at 3566, 3550 and 3523 cm$^{-1}$ are attributed to the vibrations of NH$_x$ species attached to Mo atoms, indicating the heterolysis of $H_2$ by Mo-N sites[27,48]. The evident change of kinetic isotope effects (KIE) value reflects the transformation of hydrogen-involved behavior under different driving conditions for RWGS reaction, which proves the role of photothermal catalysis to promote $H_2$ activation (Supplementary Fig. 17)[27]. Moreover, the intermediates for $CO_2$ hydrogenation can be identified under illumination, including *COOH (1287 and 1362 cm$^{-1}$), *CO (2047 and 2017 cm$^{-1}$), *CO$_3$ (1375 and 1322 cm$^{-1}$) and *CO$_2$ (1339 cm$^{-1}$). In addition, the sharply increasing signal of *COOH intermediates on $Mo_2N$/$MoO_{2-x}$ catalyst is also observed through time-of-flight secondary ion mass spectroscopy (TOF-SIMS) following the catalytic reaction (Supplementary Fig. 18). The observed DRIFTS signals of these significant intermediates indicate the pathway of $CO_2$ activation, which exhibits a rise in intensity with prolonged illumination time and aligns well with the findings of NAP-XPS results[49,50]. Based on the above spectroscopic characterizations, we propose that oxygen vacancies act as the active sites for $CO_2$ activation. In the meantime, the Mo-N sites, which receive photogenerated electrons, have the ability to rapidly dissociate $H_2$ to produce active H atoms for $CO_2$ hydrogenation due to the LSPR-induced surface electric field.

## Theoretical calculation for catalytic mechanism

With the information gleaned from spectroscopic characterizations, we further conduct DFT calculations to study the adsorption and activation process of $CO_2$ by $Mo_2N$/$MoO_{2-x}$. $Mo_2N$ and $MoO_{2-x}$ are selected as contrastive samples to investigate the synergistic effect between two separate active sites. The calculated Gibbs free energy of $H_2$ adsorption over $Mo_2N$/$MoO_{2-x}$ is −2.18 eV (Fig. 4a), suggesting that $H_2$ molecules prefer to be adsorbed on the surface of $Mo_2N$/ $MoO_{2-x}$ due to the synergistic effect at the interface[51]. It should be noted that $H_2$ molecules exhibit considerable activity on the surfaces of both $Mo_2N$/$MoO_{2-x}$ and $Mo_2N$ (Supplementary Fig. 19); however, the H-H bond should be dissociated prior to the adsorption of H atoms onto the surface. As a result, the H atoms exhibit enhanced migratory properties over $Mo_2N$/$MoO_{2-x}$, thereby having a greater propensity to establish chemical interactions with adsorbed $CO_2$ and other intermediates. On the other hand, the free energies of $CO_2$ adsorption over three model samples are −1.71, −1.31 and −2.52 eV, respectively (Fig. 4b). Interestingly, the adsorption strength of $CO_2$ on $Mo_2N$/$MoO_{2-x}$ lies at a moderate level between $Mo_2N$ and $MoO_{2-x}$. This result agrees with the Sabatier's rule, indicating that $Mo_2N$/ $MoO_{2-x}$ can serve as an efficacious catalyst[52]. The reaction free energy and the corresponding structural configuration of each intermediate are also simulated to illustrate the RWGS reaction mechanism over Mo-based catalysts. Figure 4c shows that the free energies for $CO_2$ protonation to form *COOH on $Mo_2N$/$MoO_{2-x}$, $Mo_2N$ and $MoO_{2-x}$ are −1.80, −1.16 and −2.14 eV, respectively. Based on these simulations, we infer that $MoO_{2-x}$ component is prone to drive catalytic $CO_2$ reduction and promote the formation of *COOH intermediate, which aligns with the findings from NAP-XPS. For the subsequent fundamental step, the free energies of *COOH dehydroxylation to form *CO on $Mo_2N$/$MoO_{2-x}$, $Mo_2N$ and $MoO_{2-x}$ are −4.12, −0.62 and

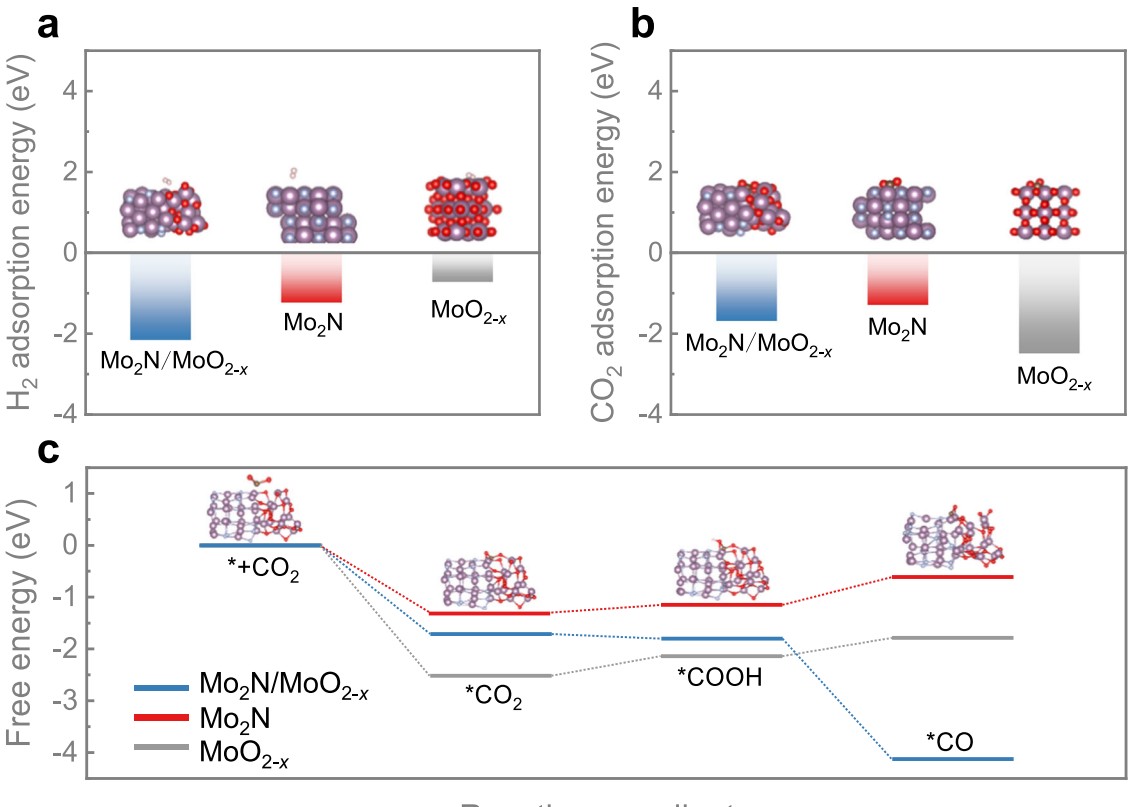

**Fig. 4 | DFT calculation for catalytic mechanism. a** Calculated $H_2$ and (**b**) $CO_2$ adsorption energy, (**c**) free energy of $Mo_2N/MoO_{2-x}$, $MoO_{2-x}$ and $Mo_2N$. The purple, red, grey, brown and champagne balls represent Mo, O, N, C, and H atoms, respectively.

−1.79 eV, respectively (Supplementary Fig. 20 and Table 4). The results demonstrate that *CO prefers to evolve on $Mo_2N/MoO_{2-x}$ catalyst, consistent with its impressive catalytic performance for photothermal RWGS reaction. Taken together, the $Mo_2N$ component in $Mo_2N/MoO_{2-x}$ is active for $H_2$ dissociation, and $CO_2$ reduction is driven by the $MoO_{2-x}$ component. The uniform distribution of $Mo_2N$ and $MoO_{2-x}$ and their sufficient combination provide adequate active sites for the synergistic activation of reactant molecules. Overall, the dual active sites of $Mo_2N/MoO_{2-x}$ catalyst exhibit a synergistic effect, together with its excellent LSPR effect, hence rendering $Mo_2N/MoO_{2-x}$ a promising photothermal catalyst for $CO_2$ hydrogenation.

In summary, we report the successful construction of a non-metallic plasmonic catalyst ($Mo_2N/MoO_{2-x}$) with dual active sites for actuating the RWGS reaction. The comprehensive characterizations and theoretical calculations demonstrate that $H_2$ and $CO_2$ can be adsorbed and activated on the N sites and oxygen vacancies simultaneously. The synergistic effect of these active sites plays a crucial role in a significant reduction of the reaction energy barrier. Meanwhile, the existence of LSPR effect in the noble-metal-free catalyst efficiently enhances the conversion of photon energy to thermal energy and optimizes the localized energy regulation, further promoting the activation of reactant molecules and facilitating the reaction. These factors result in the exceptional catalytic activity, selectivity and durability as well as energy conversion efficiency of $Mo_2N/MoO_{2-x}$ for photothermal catalysis in the RWGS reaction. This work employs a practicable design strategy to establish tunable synergistic sites with LSPR effect, and provides methodological support for understanding the mechanism of photothermal catalytic $CO_2$ hydrogenation. The energy conversion efficiency of photothermal catalysis demonstrated in our system, which has a significant advantage over thermal catalysis, will aid in advancing the development of innovative photothermal catalysts.

## Methods
### Material synthesis
To obtain $MoO_3$ nanosheets precursor, 576 mg Mo powder was dissolved in 80 mL mixed solution containing 70 mL ethanol and 10 mL $H_2O_2$. After magnetic stirring, the transparent solution was transferred into a sealed Teflon-lined autoclave under 160 °C for 24 h. The product was washed with deionized water and ethanol several times, and dried in a vacuum oven for 12 h. The as-prepared $MoO_3$ precursor was annealed in an $NH_3$ atmosphere with a flow rate of 15 SCCM (standard cubic centimeters per minute, mL·min⁻¹) for 6 h. The annealing temperatures were 450, 500, 550, 600, and 650 °C, respectively, to prepare $Mo_2N/MoO_{2-x}$ nanosheets with different proportions of $Mo_2N$ and $MoO_{2-x}$. To create oxygen vacancies in $MoO_2$ ($MoO_{2-x}$), commercial $MoO_2$ was thermally treated at 550 °C for 5 h under the Ar/$H_2$ (5%) mixture gas.

### Catalyst characterizations
Powder X-ray diffraction (XRD) patterns were recorded on a Bruker AXS D8 Advance X-ray diffractometer with a Cu Ka radiation target ($\lambda = 0.154178$ nm). Scanning electron microscopy (SEM) was conducted on Hitachi S-4800 scanning electron microscope at 5 kV. High-resolution transmission electron microscopy (HRTEM) images, energy-dispersive X-ray spectroscopy (EDS) mapping profiles and selected area electron diffraction results were collected on a JEOL-2100F system. Atomic-level high-angle annular dark-field scanning transmission electron microscopy (HAADF-STEM) images and the corresponding STEM-EDS elemental mapping profiles were taken on an FEI Titan Themis Z 3.1 equipped with a SCOR spherical aberration corrector and a monochromator. Measurements of time-of-flight secondary ion mass spectroscopy (TOF-SIMS) were conducted on a TOF-SIMS 5–100 instrument (ION-TOF GmbH). UV-vis-IR diffuse reflectance spectra were measured by a Shimadzu SolidSpec-3700

spectrophotometer in the spectral region of 200–2500 nm. A JEOL JES-FA200 electron spin resonance spectrometer was used to collect the electron paramagnetic resonance (EPR) spectra at room temperature (9.062 GHz)[53].

## Gas-adsorption analysis

Temperature-programmed desorption (TPD) measurements of reactant gases were collected by a Micromeritics AutoChem II 2920 apparatus. Catalyst powders (70 mg) were pretreated at 200 °C in Ar atmosphere for 1 h and then cooled to room temperature naturally. In order to remove other physisorbed molecules, the sample was heated at 50 °C in $H_2$/Ar (10%) mixture for 1 h and then sluiced with Ar flow. Finally, the $H_2$ desorption was measured in Ar atmosphere (30SCCM) in the temperature range from 50 °C to 600 °C with the heating rate of 10 °C min$^{-1}$. The $CO_2$ desorption was measured through the same apparatus and the similar procedures[54].

## In-situ NAP-XPS measurements

In-situ near-ambient-pressure X-ray photoelectron spectroscopy (NAP-XPS) measurements were performed on a SPECS NAP-XPS instrument, where C 1$s$, O 1$s$, N 1$s$ and Mo 3$p$ spectra were collected under UHV and 0.6 mbar $CO_2 + H_2$ (1:1) conditions. The catalyst powder was pressed onto copper foam, and then the sample was fixed onto the XPS sample holder by tantalum strips[55]. A 300 W Xenon lamp served as the white light source and was fixed outside the UHV chamber to illuminate the sample via the quartz window.

## In-situ DRIFTS measurements

In-situ diffuse reflection Fourier transform infrared spectroscopy (DRIFTS) measurements were performed through a Bruker IFS 66 v Fourier transform spectrometer equipped with a Harrick diffuse reflectance accessory at the Infrared Spectroscopy and Microspectroscopy Endstation (BL01B) of NSRL. Each spectrum was recorded by averaging 256 scans at a resolution of 4 cm$^{-1}$. The catalysts were placed in an infrared in-situ chamber sealed with ZnSe windows, which was specifically designed to examine highly scattered powder samples in diffuse reflection mode. Then the chamber was purged with Ar flow for 30 min. The spectrum was collected as a background spectrum[56]. During the in-situ characterization, $CO_2$/$H_2$ mixture gas was continually introduced into the chamber at 250 °C under visible light irradiation conditions.

## Performance evaluation of $CO_2$ hydrogenation

5 mg of catalyst was put into a quartz batch tube reactor after being loaded on a glass fiber. The quartz tube was sealed up after fulling of $CO_2$/$H_2$ mixed gas with a volume ratio of 1:1. Then the reactor was illuminated with 300 W Xe lamp (Perfect Light PLS-SEX 300D) at a specific power density (3 W·cm$^{-2}$) for 20 min. The products were detected by gas chromatography (GC-2014AF, Shimadzu) equipped with a flame ionization detector (FID). The long-term catalytic performance testing was conducted in the flow reactor under the illumination of 3 W·cm$^{-2}$, and the flow rates of hydrogen and $CO_2$ were both 10 SCCM. A K-type micro thermocouple was used to monitor the reaction temperature in real time (Supplementary Figs. 21 and 22). $CO_2$ hydrogenation reactions in thermal catalysis were tested using a homemade fixed-bed micro-reactor. 20 mg catalysts were loaded in a quartz tube with an inner diameter of 4 mm. The reactant gas consists of 24% $CO_2$ and 24% $H_2$ (volume ratio), balanced with Ar, and the flow rate was 20 SCCM. The effluent gas was online analyzed by Agilent GC6890N equipped with a TDX-1 column and thermal conductivity detector (TCD).

## Kinetic isotope effect (KIE) measurements

The KIE tests followed the same procedures as "Performance evaluation of $CO_2$ hydrogenation", replacing $H_2$ with $D_2$. The KIE value refers to the ratio of the reaction rate constant ($k$) in the $H_2$ atmosphere to that in the $D_2$ atmosphere under the same reaction conditions ($k_H/k_D$). As mentioned above, the KIE value can be directly replaced by the ratio of the CO formation rates ($r_H/r_D$) due to the high concentration of $CO_2$ and $H_2/D_2$.

## Photoelectrochemical measurements

All photoelectrochemical measurements were conducted using an electrochemical workstation (CHI 760E, Shanghai Chenhua, China) in a three-electrode system. A 300 W Xe lamp with a power density of 100 mW·cm$^{-2}$ served as the light source during the measurements. For the photoelectrode, the prepared samples were drop-coated onto FTO glass. The counter electrode was a Pt foil, while an Ag/AgCl electrode functioned as the reference electrode. These three electrodes were carefully inserted into a quartz cell, which was pre-filled with a 0.5 M $Na_2SO_4$ electrolyte solution (pH = 6.6).

## Computational details

All the calculations were performed in the framework of the density functional theory with the projector-augmented plane-wave method, as implemented in the Vienna ab initio simulation package[57]. The generalized gradient approximation proposed by Perdew, Burke, and Ernzerhof was selected for the exchange-correlation potential[58]. The long-range van der Waals interaction was described by the DFT-D3 approach[59]. The cut-off energy for the plane wave was set to 400 eV. The energy criterion was set to 10$^{-6}$ eV in the iterative solution of the Kohn-Sham equation. A vacuum layer of 15 Å was added perpendicular to the sheet to avoid artificial interaction between periodic images[60]. The k-mesh used in the calculations was chosen according to the size of the structures. $1 \times 1 \times 1$, $3 \times 3 \times 1$ and $2 \times 4 \times 1$ k-meshes were used in the calculations of $Mo_2N/MoO_{2-x}$, $Mo_2N$ and $MoO_{2-x}$, respectively. All the structures were relaxed until the residual forces on the atoms had declined to less than 0.03 eV·Å$^{-1}$.

The absorption energies of $H_2/CO_2$ on different catalysts were evaluated by the following equation:

$$E_{abs} = E_{total} - E_{slab} - E_{mol}$$

Where $E_{total}$, $E_{slab}$, and $E_{mol}$ represent to the electron's energy of the $H_2/CO_2$ absorbed on the surface of catalysts, the catalysts, and $H_2/CO_2$, respectively. Besides, the stability of all the structures with $H_2/CO_2$/intermediates absorbed on the catalysts were evaluated by the frequency calculations.

## Data availability

All data that support the findings in this paper are available within the article and its Supplementary Information or are available from the corresponding authors upon reasonable request. Source data are provided with this paper.

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

## Acknowledgements

This work was financially supported in part by the National Key R&D Program of China (2020YFA0406103), NSFC (U23A2091, 22279128, 22122506, 22075267), Jiangsu Funding Program for Excellent Post-doctoral Talent, Youth Foundation of Jiangsu Province (BK20220290), Gusu Innovation and Entrepreneurship Leading Talents Program (ZXL2022386), and Science and Technology Program of Suzhou (SWY2022003). The in-situ DRIFTS measurements were performed at beamline BL01B in the NSRL. The authors thank the support from USTC Center for Micro- and Nanoscale Research and Fabrication. This work was partially carried out at the Instruments Center for Physical Science, University of Science and Technology of China.

## Author contributions

D.L. and Y.X. supervised the project. X.W., Y.L., and Y.C. conceived and designed the experiments. X.W. performed the key experiments and analyzed the results. X.W., Y.L., Y.L., J.M., Y.L., Y.C., R.L., N.L., and K.L. assisted to carry out the in-situ EPR, DRIFTS, NAP-XPS characterization, and DFT simulations. E.Z., Y.G., Y.Z. assisted to perform the chemical synthesis. R.T.L. assisted to test the thermal catalytic performance. J.M., Y.L., D.L., and Y.X. co-wrote the manuscript. All the authors discussed the results and commented on the manuscript.

## Competing interests

The authors declare no competing interests.
