## [Peer Review File · Nature Communications]

Nonmetallic plasmonic catalyst for photothermal CO₂ flow conversion with ultrahigh activity, selectivity and durabilityReviewers' comments:

Reviewer #1 (Remarks to the Author):

This manuscript presents the utilization of Mo₂N/MoO_{2-x} nanosheets for the photothermal catalytic reverse water-gas shift (RWGS) reaction. During an extended 168-hour experiment under continuous full-spectrum light irradiation (3 W·cm⁻²), these nanosheets demonstrated a CO yield rate of 355 mmol·g⁻¹·h⁻¹ with a selectivity of 99%.

However, it is worth noting that this study can be viewed as an extension of recent research published in Nature Communications (2023, 14, 2551), which regrettably was not referenced by the authors. A comparative analysis reveals that the Ni₃N nanosheets featured in the aforementioned Nature Communications paper exhibited a catalytic activity that was at least four times greater than that of the Mo₂N/MoO_{2-x} nanosheets presented here. Consequently, the primary distinction in this work is just the substitution of Ni₃N nanosheets with Mo₂N nanosheets, albeit with poor activity.

While the catalytic investigations conducted in this study are Ok, they lack comprehensive examinations into intensity dependence, kinetic isotope effects (KIE), quantum efficiency, and related aspects. Furthermore, the claim regarding the "photo-thermal" (Vs"non-thermal") lacks substantiated proof and experimental support.

Overall, I see this MS as routine work and lacks novelty as well as in-depth mechanistic studies to be published in Nature Communication.

Reviewer #2 (Remarks to the Author):

In this manuscript, Wan et al. have designed an efficient photothermal catalyst of Mo₂N/MoO_{2-x}, which exhibits a combination of high activity and stability to catalyze the RWGS reaction under relative mild reaction conditions. The presence of localized surface plasmon resonance (LSPR) effect efficiently converts photon energy into localized heat, thus promoting the activation of H₂ and CO₂ at N atoms and O vacancies, respectively, on the catalyst surface. This work can be regarded as a good reference to guide advanced catalysts in photothermal catalysis by utilizing the LSPR effect. However, although various in-situ characterization techniques have been measured to investigate the structure-function relationship, the intrinsic structure of the catalyst and the origin of the LSPR effect are ambiguous, which largely limits the innovation and impact of this work. In summary, this manuscript needs major revisions. The specific comments are as follows:

1. Specific sizes of Mo₂N and MoO₂ are not available from HRTEM images. Please provide information about the specific size of Mo₂N and MoO₂.
2. Compared to the standard card, why the XRD peaks of Mo₂N and MoO₂ exhibit obvious shift? Please

explain it.

3. Mo₂N can be easily oxidized by O₂. Is the surface structure of the synthesized Mo₂N oxidized by O₂ in air? From the element distribution in Figure 1d, how to understand that the spatial distributions of Mo, N and O are highly correlated?
4. The authors mention that the composition ratio of MoO_{2-x} and Mo₂N can be tuned by controlling the annealing temperature. Is it possible to quantify the composition ratio of Mo₂N and MoO₂ accurately?
5. Compared to other noble metal-free catalysts, why this Mo₂N/MoO_x catalyst can generate the strong LSPR effect? And how to understand the interaction between Mo₂N and MoO_x? It seems that there is no detailed discussion in this manuscript to explain the origin of the LSPR effect over Mo₂N/MoO_x.
6. The authors emphasize that the synergy between Mo₂N and MoO_{2-x} is crucial to optimize the RWGS reaction activity. However, it seems that Mo₂N and MoO_x just serve as separate sites to activate H₂ and CO₂, respectively.
7. Although the oxygen vacancy concentrations of MNO-450, 550, and 650 are very different, their ability to adsorb CO₂ is similar (Supplementary Fig. 9). Therefore, the relationship of oxygen vacancies and CO₂ adsorption is ambiguous in this manuscript.
8. Generally, metal sites have a strong capacity to dissociate H₂ molecule. For Mo₂N/MoO_x, how to understand the role of exposed Mo sites in dissociating H₂?

Reviewer #3 (Remarks to the Author):

Liu, Xiong and coworkers report the preparation of Mo₂N/MoO_{2-x} nanosheets as catalysts for the light-powered RWGS reaction. The preparation and in-depth structural and compositional characterization of the catalyst materials produced in this study are reported very well. However, with the generation and interpretation of the obtained catalysis results I have some concerns that need to be addressed before considering publication (see below). If these concerns are addressed properly, this study could be highly significant and impact the field of photothermal plasmon catalysis.

Concerns:

- For noble metal/metal oxide combinations, you report systems with rather low activity (lines 61-67). One of the higher activities reported to date are obtained for Au/TiO₂. See e.g. Plasmon-enhanced reverse water gas shift reaction over oxide supported Au catalysts - Catalysis Science & Technology (RSC Publishing) and Low Temperature Sunlight-Powered Reduction of CO₂ to CO Using a Plasmonic Au/TiO₂ Nanocatalyst - Martínez Molina - 2021 - ChemCatChem - Wiley Online Library.
- You claim that it is necessary for the application of photothermal catalysis to move away from the use of noble metals (lines 72-75). Please note: the noble metal does not always dominate the levelized costs of the product. Recently, for photothermal CO₂ methanation it was demonstrated that the levelized cost of green H₂ was dominating the CH₄ cost price (Renewable natural gas as climate-neutral energy carrier? - ScienceDirect).
- You tested the long term stability (160 h) of you catalyst under steady state operating conditions (fig. 2d). Normally, one would expect a slight decrease of the performance over time due to water

adsorption. You don't seem to see that. Why is that the case? What are the catalyst bed temperature and reactor temperature?

-You tested the long term stability (160 h) of your catalyst under steady state operating conditions (fig. 2d). For sunlight powered chemical processes, stability of your catalyst during repetitive on/off experiments is also of importance. Did you test that?

-Figure 2f implies things that are not correct. It is misleading. You cannot simply compare the activity of your catalyst 1-to-1 to other reported catalysts, unless the test conditions are identical (e.g., same irradiance, same feedstock ratio, same flow, etc.). Your results are e.g. generated with double the irradiance of the results reported by Molina et al. (see above). Furthermore, the fact that other groups report shorter stability may also simply be attributed to the fact that they didn't test it longer than a couple of hours.

-Are you sure the broadband absorption of your catalysts is based on the LSPR effect? I know that plasmonic resonances are broad, but they typically do lead to a rather distinct absorption maximum.

-What is the highest apparent quantum efficiency you achieve (I think you call it ITC in the Supplementary Info)? This seems rather low compared to state of art systems which is not in favor of a substantial non-thermal contributor.

-How did you determine the surface temperature of your catalyst? Insufficiently careful measurements can lead to rather large underestimations (see e.g., Using Fiber Bragg Grating Sensors to Quantify Temperature Non-Uniformities in Plasmonic Catalyst Beds under Illumination - Xu - 2022 - ChemPhotoChem - Wiley Online Library). Accurate temperature measurements, incl. temperature gradient measurements, are needed to substantiate your claim on a non-thermal contributor (lines 179-182). Also other types of experiments, such as a study of the activity as function of irradiance, may contribute (see e.g., <https://www.nature.com/articles/s41377-020-00345-0>).

Reply to Reviewers' comments

Reviewer #1 (Remarks to the Author):

This manuscript presents the utilization of Mo₂N/MoO_{2-x} nanosheets for the photothermal catalytic reverse water-gas shift (RWGS) reaction. During an extended 168-hour experiment under continuous full-spectrum light irradiation (3 W·cm⁻²), these nanosheets demonstrated a CO yield rate of 355 mmol·g_{cat}⁻¹·h⁻¹ with a selectivity of 99%. However, it is worth noting that this study can be viewed as an extension of recent research published in Nature Communications (2023, 14, 2551), which regrettably was not referenced by the authors. A comparative analysis reveals that the Ni₃N nanosheets featured in the aforementioned Nature Communications paper exhibited a catalytic activity that was at least four times greater than that of the Mo₂N/MoO_{2-x} nanosheets presented here. Consequently, the primary distinction in this work is just the substitution of Ni₃N nanosheets with Mo₂N nanosheets, albeit with poor activity.

Reply: We really appreciate the reviewer's insightful suggestions to help us significantly improve the quality of our manuscript. We have carefully revised the manuscript and sincerely hope that our revisions have satisfactorily addressed the reviewer's concerns. Firstly, we are very sorry for neglecting the work of Ni₃N, which investigated the role of plasmon excitation. It can also prove the importance of developing efficient nonmetallic plasmonic catalysts for photothermal catalytic applications. However, the excellent catalytic activity of Ni₃N (1212 mmol·g_{cat}⁻¹·h⁻¹) only appeared once in the article and cannot be maintained for a long time. **As we all know, besides catalytic activity and selectivity, the stability and conversion rate are also regarded as the evaluation criteria for catalyst performance.** In long-term stability study, the catalytic activity (325 mmol·g_{cat}⁻¹·h⁻¹) of Ni₃N decreased by ~30% within 25 hours under 2.5 W·cm⁻², while the catalytic activity of Mo₂N in our work can maintain for 190 hours even under a higher light intensity (3 W·cm⁻², 355 mmol·g_{cat}⁻¹·h⁻¹) benefiting from the unique composite structure. The comparative data between Ni₃N and Mo₂N/MoO_{2-x} is shown in Table R1. It is worth noting that the reaction rate is related to many parameters, such as catalyst weight and flow rate. The CO yield rate

can reach up to $1345 \text{ mmol} \cdot \text{g}_{\text{cat}}^{-1} \cdot \text{h}^{-1}$ for the first 5 minutes in a batch reactor, which is consistent with the reaction equilibrium time in sealing system (Fig. R1/Supplementary Fig. 5). Moreover, there is also a significant difference between our catalyst and Ni_3N on the catalyst design concept. The synergistic effect of dual active sites in our study will provide a convenient avenue for the design of efficient photothermal catalysts.

Above all, we remain convinced that our work is worth reconsidered by Nature Communications.

Table R1. The comparison of catalytic activities with $\text{Mo}_2\text{N}/\text{MoO}_{2-x}$ and Ni_3N for photothermal catalytic CO_2 hydrogenation.

Catalyst	Light intensity ($\text{W} \cdot \text{cm}^{-2}$)	Flow rate (SCCM)	CO_2/H_2 ratio	CO rate ($\text{mmol} \cdot \text{g}_{\text{cat}}^{-1} \cdot \text{h}^{-1}$)	CO_2 conversion rate	Long-term Stability test	Activity decay after stability test
$\text{Mo}_2\text{N}/\text{MoO}_{2-x}$ (This work)	3	20 sccm	1:1	355	13.1%	190 h	~12%
Ni_3N	3.06	77 sccm	20:1	1212	1.2 %	/	/
	2.5	77 sccm	20:1	325	0.33%	25h	~30%

Fig. R1 and revised Supplementary Fig. 5. The production rate and selectivity of CO evolution for photothermal catalytic RWGS reaction in a batch reactor (a) by MNO-550 at different reaction times and (b) by $\text{Mo}_2\text{N}/\text{MoO}_{2-x}$ with different annealing temperature under $3 \text{ W} \cdot \text{cm}^{-2}$ light illumination for 5 min.

While the catalytic investigations conducted in this study are Ok, they lack

comprehensive examinations into intensity dependence, kinetic isotope effects (KIE), quantum efficiency, and related aspects. Furthermore, the claim regarding the "photothermal" (Vs "non-thermal") lacks substantiated proof and experimental support. Overall, I see this MS as routine work and lacks novelty as well as in-depth mechanistic studies to be published in Nature Communication.

Reply: We appreciate the concern and constructive comments raised by the reviewer; however, the reviewer may have overlooked the novelty demonstrated in this work. Our point-by-point responses are listed below:

1. We have explored the surface temperatures and CO generation rates of MNO-550 under different light intensities as shown in Supplementary Fig. 12b, and both can be enhanced with increasing light intensity. The remarkable distinction of surface temperature indicates that the Mo₂N/MoO_{2-x} nanosheets can efficiently promote charge transfer and accumulation of abundant plasmon hot electrons on the catalyst for thermal energy generation, which presents an excellent capacity for photothermal conversion.

2. As suggested by the reviewer, we have added kinetic isotope effects (KIE) tests. The D₂ is used to replace H₂ to investigate catalytic activity evaluation. The reaction rate decreases in thermal catalysis when the H₂ is replaced by D₂ (KIE = 1.35). However, the KIE value of Mo₂N/MoO_{2-x} in photothermal catalysis is 0.82 (Fig. R2/Supplementary Fig. 17). The apparent change of KIE value reflects the transformation of hydrogen-involved behavior under the different driving conditions for RWGS reaction, which proves the role of photothermal catalysis to promote H₂ activation.

Fig. R2 and revised Supplementary Fig. 17. The KIE results for thermal catalysis and photothermal catalysis.

Fig. R3 and revised Supplementary Fig. 6. The CO yield rate and selectivity of MNO-550 under the excited light with different wavebands for 5 min.

3. We have used the different wavebands of light to stimulate reactions to comprehend the function of light (a light density of $2 \text{ W}\cdot\text{cm}^{-2}$). The CO yield rate of MNO-550 positively correlates with the absorption spectra (Fig. R3/Supplementary Fig. 6). Due to the superimposed plasmonic characters of MoO_{2-x} and Mo_2N in the visible light region, the CO yield rate can reach up to $1180 \text{ mmol}\cdot\text{g}_{\text{cat}}^{-1}\cdot\text{h}^{-1}$ for 5 min under visible light excitation. Because of the insufficient energy density, almost no products can be detected in such a system under single wavelength light illumination. Therefore, we have provided the **Light Energy to Chemical Energy Conversion Efficiency (LTC)** and **Thermal Energy to Chemical Energy Conversion Efficiency (TTC)**, referring to the calculation method in the literature (Ref S1. Nat. Common. 2023, 14, 3171; Chinese. J. Catal. 2023, 49, 113-112), to evaluate the energy conversion efficiency of the catalysis system. The LTC of MNO-550 is 0.71%. Specifically, the TCC of MNO-550 in photothermal catalysis (27.6 %) is 4~5 times higher than that in thermal catalysis (5.0 %) at the same CO_2 conversion rate level, while the required

reaction temperature can be reduced by 230~300 °C in photothermal catalysis in comparison to thermal catalysis (Supplementary Table 1).

Fig. R4 and revised Supplementary Fig. 10. The CO generation rate and selectivity of MNO-550 under long-term alternating on/off light conditions ($3\text{W}\cdot\text{cm}^{-2}$).

4. To explore the "photothermal" Vs "plasmonic non-thermal" effects, we have evaluated the catalytic performance of MNO-550 in successive light on and off conditions (Fig. R4/Supplementary Fig. 10). The CO generation rate decreases sharply when the light is turned off, which is similar to the performance of Ni₃N (Nat. Commun. 2023, 14, 2551). Due to slower response rate of thermal energy compared to light energy, the ambient temperature will be maintained for a while after turning off light. We find that after switching the light off, the catalyst will not show any obvious activity in our system. Obviously, the LSPR effect has made a significant contribution.

It is worth noting that the thermal effect is also crucial for driving the RWGS reaction. In fact, it is not easy to eliminate the influence of thermal effects in a photocatalysis or photothermal system. As shown in Supplementary Fig. 12c, both the surface temperature of the sample and the generation rate of CO decrease with ice bathing. However, the performance is still better than that of thermal catalysis even at 250 °C due to the ability of photothermal conversion in Mo₂N/MoO_{2-x}.

Overall, Mo₂N/MoO_{2-x} as a novel nonmetallic plasmonic catalyst with dual active sites exhibits its superior photothermal RWGS performance. The comprehensive characterizations (FT-EXAFS spectra, XANES spectra, in-situ DRIFTS spectra, In-situ NAP-XPS spectra, KIE test, TPD test, photoelectrochemical test, TOF-SIMS spectra,

EPR spectra, and others) and theoretical calculations (FDTD, DFT) help us to in-depth understand the catalyst and the reaction process, such as the electronic and morphology structural information, plasmonic photothermal effect and the corresponding activation process of reactants. In such a unique nano-architecture, H₂ and CO₂ can be adsorbed and activated on the N sites and O vacancies simultaneously. The synergistic effect of these active sites plays a crucial role in a significant reduction of the reaction energy barrier. Meanwhile, the existence of LSPR effect in the noble-metal-free catalyst efficiently enhances the conversion of photon energy to thermal energy and optimizes the localized energy regulation, further promoting the activation of reactant molecules and facilitating the reaction. This work employs a practicable design strategy to establish tunable synergistic sites with LSPR effect and provides methodological support for understanding the mechanism of photothermal CO₂ hydrogenation.

Reviewer 2

In this manuscript, Wan et al. have designed an efficient photothermal catalyst of $\text{Mo}_2\text{N}/\text{MoO}_{2-x}$, which exhibits a combination of high activity and stability to catalyze the RWGS reaction under relative mild reaction conditions. The presence of localized surface plasmon resonance (LSPR) effect efficiently converts photon energy into localized heat, thus promoting the activation of H_2 and CO_2 at N atoms and O vacancies, respectively, on the catalyst surface. This work can be regarded as a good reference to guide advanced catalysts in photothermal catalysis by utilizing the LSPR effect. However, although various in-situ characterization techniques have been measured to investigate the structure-function relationship, the intrinsic structure of the catalyst and the origin of the LSPR effect are ambiguous, which largely limits the innovation and impact of this work. In summary, this manuscript needs major revisions.

Reply: We greatly appreciate the reviewer's positive and valuable comments. We have carefully revised the manuscript and sincerely hope that our revisions have satisfactorily addressed the reviewer's concerns.

1. Specific sizes of Mo_2N and MoO_2 are not available from HRTEM images. Please provide information about the specific size of Mo_2N and MoO_2 .

Reply: We appreciate the reviewer's valuable comments. Our samples are mainly porous nanosheets with high specific surface area and uniformly distributed elements of Mo, N and O. Therefore, conventional morphology characterization is difficult to measure the specific size of catalyst particles accurately. In order to acquire the specific size, we have complemented high-resolution TEM characterization (Supplementary Fig. 1a and Fig. R5). Based on the HRTEM image below, we can observe that the porous

nanosheets consist of well-distributed Mo₂N and MoO₂ area with 2~5 nm size.

Fig. R5. The HRTEM image of MNO-550. The orange and red dashed circular boxes represent MoO₂ and Mo₂N, respectively.

2. Compared to the standard card, why the XRD peaks of Mo₂N and MoO₂ exhibit obvious shift? Please explain it.

Reply: We appreciate the question raised by the reviewer. Compared to MoO₃ precursor, the XRD patterns of a series of MNO samples have undergone evident changes. The peak intensity decreases but peak FWHM increases significantly, implying that ammoniation treatment will reduce the crystallinity of the nanosheets, resulting in a much smaller particle size of Mo₂N and MoO_{2-x}. As shown in Fig. R5, MNO sample has short-range order and long-range disorder, which may belong to the amorphous structure characteristic. Consequently, some peaks in standard XRD patterns of Mo₂N and MoO_{2-x} disappear.

In addition, the observed shift of peak centered at ~37° is mainly caused by the variation of intensity ratio of two overlapping peaks, i.e., 37° attributed to MoO₂ (-211) crystal plane and 37.7° attributed to Mo₂N (112) crystal plane. When increasing the ammoniating temperature, the intensity of MoO₂ (-211) peak decreases but the intensity of Mo₂N (112) peak increases. As one falls another rises, the overlapped peak exhibits a slight shift toward a higher diffraction angle.

3. Mo₂N can be easily oxidized by O₂. Is the surface structure of the synthesized Mo₂N

oxidized by O₂ in air? From the element distribution in Fig. 1d, how to understand that the spatial distributions of Mo, N and O are highly correlated?

Reply: We appreciate the questions raised by the reviewer. After ammoniation of MoO₃ precursor, the as-prepared sample will quickly be partially oxidized upon removal from the tube furnace and exposure to air. Similar phenomenon has been reported by Kim et al. (Ref. 25 Appl. Surf. Sci. 1999, 152, 35-43). For the highly correlated spatial distribution of Mo, N and O shown in the Fig. 1d, we try to understand from two aspects. On one hand, during the catalyst synthesis process, the porous nanosheets with high specific surface area facilitate full contact between the surface of MoO₃ precursor and NH₃ reactant. As a result, the distribution of N and O in the catalyst is relatively uniform. On the other hand, the low NH₃ flow rate (<10 SCCM) and low heating rate (1 °C/min) lead to the slow ammoniation of MoO₃, and the O atoms in the lattice are gradually replaced by N atoms. The formation of Mo₂N is limited by growth kinetics and annealing temperature, and consequently, MoO₃ cannot be completely ammoniated to Mo₂N (J. Phys. Chem. C 2010, 114, 14710–14715).

4. The authors mention that the composition ratio of MoO_{2-x} and Mo₂N can be tuned by controlling the annealing temperature. Is it possible to quantify the composition ratio of Mo₂N and MoO₂ accurately?

Reply: We thank the reviewer for his/her valuable question. N and O are two kinds of light atoms with close atomic radii while Mo has abundant valence states, which make it hard to distinguish MoO_{2-x} and Mo₂N for accurate quantification.

XPS, as a semi-quantitative characterization technique, can partially reflect the variation pattern of different component ratios. Here, we have performed peak fitting on Mo 3p XPS spectra of different samples, shown as Fig. R6 and Table R2. Each sample contains an evident MoO₃ substrate phase. The ratio of Mo₂N to MoO₂ increases from 0.22 (MNO-450) to 0.68 (MNO-650) with the increasing ammoniation temperature, in line with expectations.

Fig. R6 and revised Fig. 1f. Mo 3p and N 1s XPS spectra of Mo₂N, MoO_{2-x} and Mo₂N/MoO_{2-x} samples at different annealing temperatures. The orange, blue, green and magenta peaks are attributed to MoO₃, MoO₂, Mo 3p and N 1s of Mo₂N components, respectively.

Table R2. The ratio of Mo₂N to MoO₂ calculated by peak fitting of Mo 3p.

Sample	MoO _{2-x}	MNO-450	MNO-550	MNO-650	Mo ₂ N
ratio of Mo ₂ N to MoO ₂	0	0.22	0.60	0.68	0.80

5. Compared to other noble metal-free catalysts, why this Mo₂N/MoO_x catalyst can generate the strong LSPR effect? And how to understand the interaction between Mo₂N and MoO_x? It seems that there is no detailed discussion in this manuscript to explain the origin of the LSPR effect over Mo₂N/MoO_x.

Reply: We appreciate the reviewer's valuable comments. The LSPR effect in Mo₂N/MoO_{2-x} nanosheets occurs from the collective oscillation of conduction electrons at the surface of the nanoparticles under the excitation of incident light. The interaction of LSPR between Mo₂N and MoO_{2-x} is related to the microstructure of the catalyst, which has been replied in the comments above. There is a large number of interfaces between Mo₂N and MoO_{2-x} components in Mo₂N/MoO_{2-x} nanosheets and the interfacial

interaction can regulate the strength of LSPR effect. Both Mo₂N and MoO₂ components are reported as nonmetallic plasmonic materials in the previous literatures (Nano Lett. 2021, 21, 4410-4414; Appl. Catal. B 2022, 319, 121887). Mo₂N component can enhance the absorbance of catalyst in the vis-NIR region (Fig. 2a), while the oxygen vacancies in MoO_{2-x} can capture the photogenerated electrons to promote charge separation. The existence of interfaces can enhance photogenerated charge transfer and the accumulation of hot carriers can increase the surface temperature of catalyst, thus promoting reactions in our system (Small 2023, 16, 2301280; Energy Environ. Sci. 2023, 16, 3462-3473).

Based on previous reports, we have conducted the following experiments to demonstrate the existence and importance of the LSPR effect in this system:

1. The results of UV-vis-NIR absorption spectra show that Mo₂N/MoO_{2-x} nanosheets exhibit a significant absorption enhancement in the vis-NIR region. The plasmonic peaks, affected by varying catalyst grain sizes (Fig. 2a and Supplementary. Fig. 2), would overlap and cover each other, resulting a broad absorption range. Similar phenomena can also be observed in TiN (Nano Energy, 2022, 104, 107989).

2. We have found that the surface temperatures and CO generation rates of MNO-550 are promoted with increasing light intensity (Supplementary. Fig. 12b). The remarkable distinction of surface temperature indicates that the hot electrons from LSPR effect can accumulate on the catalyst for thermal energy generation, which presents an excellent photothermal conversion capacity of Mo₂N/MoO_{2-x} nanosheets (Adv. Mater. 2022, 34, 2202367).

3. The finite-difference time-domain (FDTD) simulations can also confirm the existence of strong LSPR effect by the local electric field enhancement at the interface of Mo₂N and MoO_{2-x} (Fig. 2b).

6. *The authors emphasize that the synergy between Mo₂N and MoO_{2-x} is crucial to optimize the RWGS reaction activity. However, it seems that Mo₂N and MoO_x just serve as separate sites to activate H₂ and CO₂, respectively.*

Reply: We appreciate the questions raised by the reviewer. The individual activities of

Mo₂N and MoO_{2-x} are not high as shown in Fig. 2c. Based on the reviewer's suggestions, we mix Mo₂N and MoO_{2-x} powders physically and mechanically in different proportions and then test their photothermal CO₂ hydrogenation performance. As shown in Fig. R7 (Supplementary Fig. 7), the activities of all mixed samples are below 3 mmol·g_{cat}⁻¹·h⁻¹, much lower than the activity of MNO-550 (385 mmol·g_{cat}⁻¹·h⁻¹). Moreover, our theoretical calculation shows that the free energy of *COOH dehydroxylation to form *CO on Mo₂N/MoO_{2-x} is much lower than that on Mo₂N and MoO_{2-x} surface. The comprehensive characterizations and theoretical calculations in our manuscript demonstrate that H₂ and CO₂ can be adsorbed and activated on the N sites and O vacancies simultaneously. Efficient CO₂ hydrogenation demands the coordinating activation of H₂ and CO₂ (Nat. Commun. 2021, 12, 3884; ACS Energy Lett. 2021, 6, 2024-2029). In our work, we have achieved the outstanding photothermal RWGS performance results from the synergistic effect of uniformly distributed Mo₂N and MoO_{2-x} in the catalyst. Combining previous literature and our experimental and theoretical results, we can conclude that Mo₂N and MoO_{2-x} do not just serve as active sites for H₂ and CO₂ activation separately. By regulating the interaction between two components, we could enhance the synergistic effect of Mo₂N and MoO_{2-x} through a facile one-step annealing method, fully utilizing their respective advantages, which simultaneously achieve high activity, high selectivity, and long-term stability.

Fig. R7 and revised Supplementary Fig. 7. The production rate and selectivity of CO evolution for physically mixed Mo₂N and MoO_{2-x} catalysts under 3 W·cm⁻² full-spectrum light irradiation. Data of MNO-550 are also listed for comparison.

7. Although the oxygen vacancy concentrations of MNO-450, 550, and 650 are very different, their ability to adsorb CO₂ is similar (Supplementary Fig. 14). Therefore, the relationship of oxygen vacancies and CO₂ adsorption is ambiguous in this manuscript.

Reply: We appreciate the reviewer's valuable comments. The difference between CO₂-TPD peak of our samples is indeed not significant enough. The intensities of peaks at 531.5 eV can be attributed to V_o as shown in the O 1s XPS spectra of different samples in Fig. 3b. It is not difficult to comprehend that the concentration of oxygen vacancies is determined by the annealing temperature as higher annealing temperature leads to a more sufficient ammonization reaction and fewer oxygen vacancies. In order to seek more convincing evidence of the relationship between oxygen vacancies and CO₂ adsorption, we have performed NAP-XPS to examine the adsorption of CO₂ on different MNO samples in the dark. Upon feeding CO₂, the O 1s XPS peaks results are shown as Fig. R8. The higher binding energy in MoO_x (brown) signifies a more vital interfacial interaction among the nano-architecture with rising annealing temperatures. We can distinctly observe that the peak intensity of adsorbed CO₂ (magenta), oxygen vacancy (blue) and metal oxide (brown) all decrease monotonously with increasing annealing temperatures. We can introduce more oxygen vacancies and strengthen CO₂ adsorption by raising the ammonization annealing temperature. According to previous research on MoO₂ for solar-driven CO₂ conversion (Angew. Chem. Int. Ed. 2023, 62, e202213124; J. Mater. Chem. A 2021, 9, 13898), the adsorption of CO₂ highly depends on the oxygen vacancy. Therefore, combined with the prior results, these findings allow us to conclude that CO₂ adsorption capacity is contingent upon oxygen vacancy concentration in our samples.

Fig. R8 and revised Supplementary Fig. 15. In-situ O 1s NAP-XPS spectra for various samples in 0.3 mbar CO₂ without illumination. The magenta, blue, and brown peaks are associated with oxygen originating from adsorbed CO₂, oxygen vacancy and metal oxide, respectively.

8. Generally, metal sites have a strong capacity to dissociate H₂ molecule. For Mo₂N/MoO_x, how to understand the role of exposed Mo sites in dissociating H₂?

Reply: We appreciate the reviewer's valuable comments. Here, we should divide the discussion into two aspects, i.e., Mo sites on Mo₂N and Mo sites on MoO_x. In Mo₂N, the adsorption and heterolysis of H₂ molecules occur on the surface Mo-N sites owing to the strong noble metallic properties of Mo₂N. Metallic character of Mo₂N has been demonstrated by its continuous distribution of partial density of states (DOS) near the Fermi level (Electrochimi. Acta 2020, 364, 137219). The exposed Mo sites play a role in the activation and dissociation of H₂ molecules (ACS Energy Lett. 2021, 6, 2024-2029). As for MoO_x, the variable valence states of Mo among Mo⁶⁺, Mo⁵⁺ and Mo⁴⁺ provide a suitable environment for electron transfer between Mo and O sites (Research 2021, 5130420; Green Chem. 2021, 23, 7259), which is conducive to form oxygen vacancies for CO₂ activation on the MoO_x surface.

Reviewer 3

Liu, Xiong and coworkers report the preparation of Mo₂N/MoO_{2-x} nanosheets as catalysts for the light-powered RWGS reaction. The preparation and in-depth structural and compositional characterization of the catalyst materials produced in this study are reported very well. However, with the generation and interpretation of the obtained catalysis results I have some concerns that need to be addressed before considering publication (see below). If these concerns are addressed properly, this study could be highly significant and impact the field of photothermal plasmon catalysis.

Reply: We greatly appreciate the reviewer's highly positive rate and valuable comments. Our point-by-point responses are listed below.

1. For noble metal/metal oxide combinations, you report systems with rather low activity (lines 61-67). One of the higher activities reported to date are obtained for Au/TiO₂. See e.g. Plasmon-enhanced reverse water gas shift reaction over oxide supported Au catalysts - Catalysis Science & Technology (RSC Publishing) and Low Temperature Sunlight - Powered Reduction of CO₂ to CO Using a Plasmonic Au/TiO₂ Nanocatalyst - Martínez Molina - 2021 - ChemCatChem - Wiley Online Library.

Reply: We thank the reviewer for his/her valuable comments. These two works can indeed represent noble metal/metal oxide plasmonic nanostructures for CO₂ conversion. In the first article (Catal. Sci. Technol. 2015, 5, 2590-2601), the authors reported that an oxide-supported Au catalysts showed 30 to 1300% higher activity for RWGS under visible light compared to dark conditions. Their kinetic results indicated that LSPR effect increases the rate of hydroxyl hydrogenation and carboxyl decomposition. In the second work (ChemCatChem 2021, 13, 4507-4513), the authors demonstrated that Au/TiO₂ is able to selectively promote the RWGS reaction under slightly concentrated artificial sunlight illumination without any external heating source. Moreover, they found that light helps to efficiently promote CO formation and suppresses the methanation reaction. The above two works both illustrate the advantages of photothermal catalysis in RWGS and the contribution of LSPR in the reaction. We have added relevant literature in the revised manuscript:

"For example, Au/TiO₂ is one typical photocatalyst with LSPR effect.¹⁸⁻²⁰ Fan et al. proved that hot electrons generated by LSPR can promote the formation of oxygen vacancies in Au/TiO₂ catalyst, facilitating the adsorption and activation of CO₂.¹⁹ Sastre et al. demonstrated that plasmonic Au/TiO₂ nanostructure could drive photothermal RWGS reaction with a CO generation rate of 429 mmol g_{Au}⁻¹·h⁻¹ (13.4 mmol·g_{cat}⁻¹·h⁻¹) under 14.4 sun irradiation.²⁰"

2. You claim that it is necessary for the application of photothermal catalysis to move away from the use of noble metals (lines 72-75). Please note: the noble metal does not always dominate the levelized costs of the product. Recently, for photothermal CO₂ methanation it was demonstrated that the levelized cost of green H₂ was dominating the CH₄ cost price (Renewable natural gas as climate-neutral energy carrier? - ScienceDirect).

Reply: We appreciate the reviewer's valuable comments. We agree with the reviewer's comments that H₂ dominates the industrial cost of CO₂ hydrogenation. However, we still believe that developing efficient catalysts and reducing the use of noble metals are extremely necessary. Moreover, it is significative to conduct an in-depth investigation of the LSPR effect on a non-noble metal catalyst. We have revised the manuscript as follows: "However, these catalysts still require the involvement of noble metals to activate H₂ and boost their catalytic activity, which increases the intricacy of the catalyst synthesis process".

3. You tested the long term stability (160 h) of you catalyst under steady state operating conditions (Fig. 2d). Normally, one would expect a slight decrease of the performance over time due to water adsorption. You don't seem to see that. Why is that the case? What are the catalyst bed temperature and reactor temperature?

Reply: We thank the reviewer for his/her constructive suggestions. We have extended the stability testing time to 190 hours, and a slight decrease of the performance can be observed at 190th hour (12%). The stability results have been updated in the revised manuscript as Fig. 2d. During the reaction process, the surface temperature of the

catalyst can reach 250 °C, while the temperature measured from the bottom of the reactor is about 65 °C.

Fig. R9 and revised Fig. 2d. Long-term stability test of MNO-550 in a flow reactor.

4. You tested the long term stability (160 h) of your catalyst under steady state operating conditions (Fig. 2d). For sunlight powered chemical processes, stability of your catalyst during repetitive on/off experiments is also of importance. Did you test that?

Reply: We thank the reviewer for his/her thoughtful suggestions. We have supplemented the repetitive on/off experiment, and the result is shown in Fig. R10 and revised Supplementary Fig. 10. Our catalyst can maintain its remarkable activity after 220 hours light on-off reaction.

Fig. R10 and revised Supplementary Fig. 10. Long-term stability test of MNO-550 for repetitive on/off experiment.

5. Fig. 2f implies things that are not correct. It is misleading. You cannot simply compare the activity of your catalyst 1-to-1 to other reported catalysts, unless the test conditions are identical (e.g., same irradiance, same feedstock ratio, same flow, etc.). Your results are e.g. generated with double the irradiance of the results reported by Molina et al. (see above). Furthermore, the fact that other groups report shorter stability may also simply be attributed to the fact that they didn't test it longer than a couple of hours.

Reply: We thank the reviewer for his/her valuable comments. Due to the various reaction conditions used by different research groups in photothermal catalysis, it is difficult to compare the activity under identical conditions. In order to increase the clarity and reliability of activity comparison, we have summarized and listed the specific testing conditions for the photocatalytic activity in different works in SI, shown as Table R3 (**Supplementary Table 2**). Our catalyst activity is still leading among these catalysts under the same lighting and atmosphere conditions. Molina et al. reported that the Au/TiO₂ activity is 13.4 mmol·g_{cat}⁻¹·h⁻¹ at 1.4 W·cm⁻², and our MNO-550's activity can reach 18 mmol·g_{cat}⁻¹·h⁻¹ under 1 W·cm⁻² light intensity, which is still slightly higher than their experimental results.

Supplementary Table 2. The specific testing conditions and the photothermal activities of Mo₂N/MoO_{2-x} and state-of-the-art catalysts.

Catalyst	Light intensity	External heating temperature	CO ₂ /H ₂ ratio	CO yield rate (mmol·g _{cat} ⁻¹ ·h ⁻¹)	Long-term stability test	Activity decay after stability test
Mo ₂ N/MoO _{2-x} (This work)	3 W·cm ⁻²	/	1:1	1345 (reaction for 5min)	/	/
	1 W·cm ⁻²	/	1:1	18.1	/	/
	2 W·cm ⁻²	/	1:1	264	/	/
	3 W·cm⁻²	/	1:1	355	190	~12%
Ni ₃ N	4 W·cm ⁻²	/	1:1	454	/	/
	3.06 W·cm ⁻²	/	20:1	1212	/	
	2.5 W·cm ⁻²	/	20:1	325	25	~30%
Ru/Mo ₂ TiC ₂	3.4 W·cm ⁻²	/	1:1	312	15	~3%
Au/TiO ₂	1.4 W·cm ⁻²	/	1:1	13.4	4	/
Au/TiO ₂ (DP)	0.52 W·cm ⁻²	400 °C	1:2	159.8	/	/
Ga-Cu/CeO ₂	3.82 W·cm ⁻²	/	1:1	337.2	10	~28%
CF-Cu ₂ O	4 W·cm ⁻²	/	5:1	139.6	60	~13 %
Ni@SiO ₂	2.8 W·cm ⁻²	/	1:1	44.1	35	~2%
Cu/2D-Si	3.4 W·cm ⁻²	/	1:4	13	5	/
Pd@Nb ₂ O ₅	4.2 W·cm ⁻²	/	1:1	18.8	3.5	/
Ni ₁₂ P ₅ /SiO ₂	0.8 W·cm ⁻²	/	5:1	13.5	100	/
	2.3 W·cm ⁻²	/	5:1	960	/	/
TiN@TiO ₂	1.6 W·cm ⁻²	300 °C	1:3	13.04	40	~10%
Fe ₃ O ₄	2.05 W·cm ⁻²	/	1:2	11.3	24	/
Mo ₂ NH _x	0.35 W·cm ⁻²	175 °C	1:4	0.35	15	/
Black In ₂ O ₃	2..0 W·cm ⁻²	/	1:1	1.87	70	~6%
CuNi/CeO ₂	Visible light	310 °C	1:4	1.3	25	/
Pt/H _x MoO _{3-y}	Visible-infrared	140 °C	1:1	1.2	6	/

* "~" refers to the approximation for each reference according to their provided mass and reaction time of catalysts in the experimental section.

6. Are you sure the broadband absorption of your catalysts is based on the LSPR effect? I know that plasmonic resonances are broad, but they typically do lead to a rather distinct absorption maximum.

Reply: We thank the reviewer for the careful review. In fact, the similar broadband absorption caused by LSPR effect has been generally observed in TiN (Nano energy, 2022, 104, 107989; Sci. Rep. 2019, 9, 15287). The plasmonic resonance is determined by the size, shape and material of the nanoparticle. The size distribution of Mo₂N/MoO_{2-x} nanosheets widely ranges from 20 to 250 nm. Due to the discrepancy of their grain size, the plasmonic peaks of Mo₂N/MoO_{2-x} nanosheets would overlap and cover each other. Consequently, the maximum absorption peak cannot be displayed significantly (Fig. R11). Moreover, the LSPR effect in Mo₂N/MoO_{2-x} nanosheet can be confirmed by the finite-difference time-domain (FDTD) simulations and the changes of surface temperatures of Mo₂N/MoO_{2-x} under different light intensities, which is consistent with previous literature (Energy Environ. Sci. 2023, 16, 3462).

Fig. R11 and revised Supplementary Fig. 2. (a) UV-vis-NIR absorption spectra of Mo-based catalysts. (b) Grain size distribution of Mo₂N/MoO_{2-x} nanosheets.

7. What is the highest apparent quantum efficiency you achieve (I think you call it ITC in the Supplementary Info)? This seems rather low compared to state of art systems which is not in favor of a substantial non-thermal contributor.

Reply: We apologize for the misunderstanding caused by our description to the reviewers. **Input Energy to Chemical Energy Conversion Efficiency (ITC)** calculation of thermal RWGS reaction is used to evaluate the efficiency of energy

conversion for thermal catalysis. Due to the severe thermal diffusion in thermal catalysis, the ITC efficiency is very low. This is not the same concept as the apparent quantum efficiency described by the reviewer. This ITC parameter is generally not chosen to be reported in the laboratory because of the thermal diffusion loss. However, in the industrial production process, there will be waste heat recovery and utilization devices to improve ITC efficiency. Here, we define the efficiency of light energy to chemical energy using **Light Energy to Chemical Energy Conversion Efficiency (LTC)**, referring to the calculation method in the literature (Nat. Common. 2023, 14, 317), with an efficiency of 0.82%. We have also calculated the **Thermal Energy to Chemical Energy Conversion Efficiency (TTC)** to eliminate the interference of environmental thermal diffusion (Chin. J. Catal. 2023, 49, 113-112). By comparing LTC, ITC and TTC, the advantages of photothermal catalysis in thermal management are fully demonstrated (Supplementary Table 1).

8. How did you determine the surface temperature of your catalyst? Insufficiently careful measurements can lead to rather large underestimations (see e.g., Using Fiber Bragg Grating Sensors to Quantify Temperature Non - Uniformities in Plasmonic Catalyst Beds under Illumination - Xu - 2022 - ChemPhotoChem - Wiley Online Library). Accurate temperature measurements, incl. temperature gradient measurements, are needed to substantiate your claim on a non-thermal contributor (lines 179-182). Also other types of experiments, such as a study of the activity as function of irradiance, may contribute (see e.g., <https://www.nature.com/articles/s41377-020-00345-0>).

Reply: We thank the reviewer for his/her kind suggestions. We used the thermocouples to measure the real-time reaction temperature, the same as previous literature (Nat. Common. 2023, 14, 2551; Nat. Common. 2020, 11, 5149 and ACS Energy Lett. 2021, 6, 2024). We used copper foil with a high thermal conductivity as the substrate to load sample to ensure the accuracy and reliability of the measured temperature (Fig. R12). As shown in Fig. R13, the real-time temperature rise curves of Cu foil without catalyst and thermocouple directly irradiated by the light source are basically identical (~70 °C).

It proves the reliability of this method. However, when loading the catalyst, the real-time sample temperature can quickly rise to 250 °C, just as described in our manuscript.

Fig. R12 and revised Supplementary Fig. 21. Schematic diagram of temperature measurement for reaction device.

Fig. R13 and revised Supplementary Fig. 22. The corresponding temperature rise curves of catalyst on Cu foil, pure Cu foil and only thermocouple.

REVIEWERS' COMMENTS

Reviewer #1 (Remarks to the Author):

I would like to express my appreciation for the authors' meticulous attention to my earlier comments, specifically regarding the comparison between their catalysts and Ni₃N. The authors have now thoroughly addressed this aspect, convincingly demonstrating that their catalysts exhibit superior stability compared to Ni₃N. Indeed, stability is a critical parameter in catalysis.

Furthermore, I concur with the authors' assertion that the reaction rate is influenced by various factors, including light intensity, catalyst weight and flow rate. Consequently, a direct comparison between Ni₃N and the Mo₂N/MoO_{2-x} nanosheets may not yield precise insights, as clearly evidenced by their data in Table R1.

Moreover, the authors have adeptly addressed my previous concerns regarding mechanistic studies. The inclusion of comprehensive kinetic isotope effect studies, wavelength-dependent investigations, and light on-off experiments in this revision, elucidated the thermal and non-thermal effects in their system.

In light of these comprehensive revisions, I recommend its publication in Nature Communications as-is.

Reviewer #2 (Remarks to the Author):

The authors have modified the manuscript with care and responded to the reviewers' comments in detail. I suggest accepting it.

Reviewer #3 (Remarks to the Author):

The authors have carefully and thoroughly addressed all my concerns. I have no further objections against publication.

Response to referees

Reviewer #1 (Remarks to the Author):

I would like to express my appreciation for the authors' meticulous attention to my earlier comments, specifically regarding the comparison between their catalysts and Ni₃N. The authors have now thoroughly addressed this aspect, convincingly demonstrating that their catalysts exhibit superior stability compared to Ni₃N. Indeed, stability is a critical parameter in catalysis.

Furthermore, I concur with the authors' assertion that the reaction rate is influenced by various factors, including light intensity, catalyst weight and flow rate. Consequently, a direct comparison between Ni₃N and the Mo₂N/MoO_{2-x} nanosheets may not yield precise insights, as clearly evidenced by their data in Table R1.

Moreover, the authors have adeptly addressed my previous concerns regarding mechanistic studies. The inclusion of comprehensive kinetic isotope effect studies, wavelength-dependent investigations, and light on-off experiments in this revision, elucidated the thermal and non-thermal effects in their system.

In light of these comprehensive revisions, I recommend its publication in Nature Communications as-is.

Reply: We really appreciate the reviewer for his/her previous review and the recent feedback which definitely help us to significantly improve the quality of our manuscript.

Reviewer #2 (Remarks to the Author):

The authors have modified the manuscript with care and responded to the reviewers' comments in detail. I suggest accepting it.

Reply: We sincerely appreciate the reviewer for the positive and constructive comments regarding our manuscript.

Reviewer #3 (Remarks to the Author):

The authors have carefully and thoroughly addressed all my concerns. I have no further objections against publication.

Reply: We thank the reviewer for his/her professional review work, constructive comments, and valuable suggestions on our manuscript.